# The Role of the Gut Microbiome and Probiotics in Sports Performance: A Narrative Review Update

**DOI:** 10.3390/nu17040690

**Published:** 2025-02-14

**Authors:** Harry Jarrett, Sophie Medlin, James C. Morehen

**Affiliations:** 1Department for Research and Development, Heights, London W1D 2LG, UK; sophie@citydietitians.co.uk; 2City Dietitians, London WC2E 7PP, UK; 3Research Institute for Sport and Exercise Sciences, Liverpool John Moores University, Liverpool L3 2EX, UK; morehenperformanceltd@gmail.com

**Keywords:** probiotics, gut microbiome, exercise performance, nutrient absorption, immune system

## Abstract

**Background/Objectives:** Gut microbiome modulation through probiotics is a growing area of research, with several investigations reporting beneficial health outcomes for the host. Physical exercise has been shown to impact gut microbiome diversity. Emerging evidence suggests that probiotic supplementation can affect exercise performance. However, the mechanisms and domain-specific effects of gut microbiome modulation on performance remain to be elucidated. This narrative review aims to investigate the potential mechanisms underpinning the ergogenic benefits of probiotics and further define the current evidence base for specific performance domains. **Discussion**: The literature suggests that improved recovery after intense training regimes, enhanced nutrient absorption, alleviation of gastrointestinal symptoms, and improved immune function may underpin the beneficial effects of probiotics on sporting performance. A small number of trials also suggest that probiotic supplementation may improve symptoms of performance anxiety. However, further research is warranted on this topic. The evidence is most substantial for improvements in endurance performance, whilst only a few trials have investigated the impact upon power performance, albeit with promising results. **Conclusions/Future Perspectives**: In summary, probiotic supplementation has been shown to improve sporting performance; future research may wish to further explore the impact on power performance and investigate specific mechanisms of action.

## 1. Introduction

“All disease begins in the gut”—Hippocrates of Kos (c. 460–c. 370 BCE), the father of modern medicine, made this statement over 2000 years ago.

The gut microbiome is the collective genomes of microbiota residing in the gut, with approximately 40 trillion microbial cells. Recent technological advances in microbiome bioinformatics and sequencing techniques have allowed science to investigate the role of the gut microbiome in human health and disease [1]. Perturbations in the gut microbiome have been linked to a plethora of different disease states, including cognitive impairment, depression, anxiety, and autism spectrum disorder, to name but a few [2]. However, gut microbiome modulation through probiotic intervention has been reported to improve a number of health outcomes significantly. For example, a recent systematic review concluded that 9 of 10 studies reported that probiotic supplementation improved cognitive function in healthy or cognitively impaired older adults [3].

The role of the gut microbiome in human health and disease is rapidly evolving, with greater evidence accumulating year on year. Emerging evidence suggests that the gut microbiome may significantly impact various aspects of sporting performance. Several randomised controlled trials (RCTs) have recently reported significant improvements in athletic performance following probiotic intervention. For example, one study reported a significant 6% increase in aerobic capacity following a 6-week probiotic intervention in healthy adults [4]. Additionally, a number of studies have found significant improvements in endurance performance across various different sporting modalities such as running, cycling, and swimming following probiotic administration [5,6,7]. Although widely overlooked, a small number of studies have also found improvements in power-specific sporting domains, such as increases in one-rep max [8] and mean power output [9], following probiotic supplementation. However, to our knowledge, no review has summarised the current state of the evidence regarding the role of probiotics in power-specific exercise domains.

The mechanisms by which probiotics may impact sporting performance are still to be elucidated. A number of proposed mechanisms have been suggested in the literature, including enhanced exercise recovery [10], improved nutrient absorption [11], and reductions in gastrointestinal symptoms [12]. As the evidence base for the mechanisms of the ergogenic benefits of probiotics on sports grows, it is time for an overarching review to summarise the current state of the evidence to help guide further research.

This narrative review aims to summarise the key evidence investigating the efficacy of gut microbiome modulation by probiotics to influence sporting performance, including power-specific sporting domains. A further aim is to provide an overview of the proposed mechanisms of action by which probiotics may enhance sporting performance.

## 2. The Gut Microbiome and Exercise

There are many factors that have been reported to influence gut microbiome diversity and subsequently may impact upon the host’s health. For example, clinical interventions such as antibiotic use, dietary intake, and smoking have all been reported to influence the gut microbiome diversity [13]. More recently, exercise has also been implicated as a contributing factor in determining the gut microbiome diversity. Specifically, several observational studies have reported that athletes have a greater gut microbial diversity than those who are sedentary [14,15]. However, there is an inverse relationship between exercise and gut health. Specifically, moderate-intensity exercise has been associated with improvements in gut microbial diversity. However, sustained high-intensity exercise (as undertaken by elite athletes) significantly reduces microbial diversity, believed to be a result of hypoperfusion [16] and a disturbed immune system, which results in an increased inflammatory response [17]. Such factors may partly underpin the adverse health outcomes associated with such intense training regimes, resulting in training and performance decrements. For example, in one study of ultramarathon runners, 96% experienced gastrointestinal (GI) symptoms, and 36% of non-finishers reported that these GI symptoms caused them to terminate the race early [5].

The research to date has primarily focused on two specific aspects of sporting performance concerning the ergogenic effects of probiotic supplementation: endurance and power performance. Below, we highlight the evidence to date reporting on the impact of probiotics on these performance outcomes.

## 3. Probiotics and Sporting Performance

### 3.1. Endurance Domains

Several RCTs have reported significant improvements in aerobic capacity and performance in various sporting domains following probiotic supplementation [7,10,18]. For example, in triathletes, a significant improvement in running performance was reported following 4 weeks of supplementation with a probiotic intervention compared to placebo [9]. At the same time, a recent study reported that a 5-week intervention with a probiotic (combined Lactobacillus and Bifidobacterium strains at 10 billion CFU in total) in marathon runners significantly improved the distance covered in the Cooper test [19]. In contrast, no change in the placebo group was observed. Interestingly, the authors also found significantly greater post-exercise muscle microperfusion as measured by functional magnetic resonance imaging (fMRI) in those assigned to the probiotic group. Such findings indicate that probiotic intervention may improve aerobic exercise efficiency through improved vascular function. Indeed, several reports have found a reduced risk of cardiovascular disease in those who regularly consume probiotics, an effect partly mediated by increased nitric oxide bioavailability and, subsequently, improvements in vascular function [20]. In addition, the authors also reported significant improvements in mood post-intervention, suggesting that probiotics may also help support mental well-being in athletes (as discussed below).

The ergogenic benefits of probiotics have also been reported to extend beyond merely running performance. For example, Mazur-Kurach et al. [7] reported that 4 months of probiotic supplementation led to significant increases in the duration of exercise to failure and a 5% increase in the $\dot{V}o_{2max}$ with a concurrent feeling of less discomfort during the endurance exercise test in cyclists. Meanwhile, an 8-week probiotic intervention in female endurance swimmers led to significant improvements in $\dot{V}o_{2max}$ compared to placebo [6]. In addition to an improved $\dot{V}o_{2max}$, a significant reduction in respiratory tract infections was reported alongside reductions in subsequent symptoms, including dyspnoea and ear pain.

Research on the influence of probiotic supplementation on endurance performance in team-based sports is limited. However, one study in young, healthy badminton players investigated the ergogenic benefits of a 6-week probiotic intervention on aerobic capacity and mental well-being [4]. A significant 6% increase in aerobic capacity in those assigned to the active probiotic group was reported compared to those receiving placebo. Moreover, the authors also reported a significant 16% and 20% reduction in anxiety and stress, respectively.

In summary, several RCTs have reported a beneficial effect of probiotic supplementation on aerobic capacity and endurance performance. The ergogenic benefits extend to various sporting domains.

### 3.2. Power Domains

In a sporting context, power is the ability to generate force over a certain period, usually over short durations. Power is a significant predictor of performance across many sporting domains, including weightlifting, track events, wrestling, and team sports like basketball, football, and rugby, to name but a few. A major determinant of power in sports is muscular strength. The role of probiotics in maintaining or improving muscle strength and muscle mass in the general and clinical population has been well documented. For example, a recent meta-analysis concluded that probiotic interventions significantly improved global muscle strength and mass [21]. In a subgroup analysis, probiotics were most efficacious when the intervention was of a duration of 12 weeks, specifically with Bifidobacterium strains. However, limited evidence has shown probiotic interventions’ roles in athletes’ power performance. Below, we provide a synthesis of the current evidence base.

In an 8-week probiotic intervention in triathletes, cycling performance was significantly improved, with specific increases in the mean power output and fatigue index (as measured by the power output during 30 s of cycling), compared to placebo [9]. Meanwhile, an intervention with probiotics and combined protein for 60 days in resistance-trained males led to significant increases in the one-rep max and vertical jump power by 17% and 8%, respectively, compared to those consuming protein alone [8]. Similarly, Jäger et al. compared the influence of a combined probiotic and protein intervention to protein alone on power performance (Wingate peak power) and muscle recovery following a fatigue-inducing exercise regime [12]. The authors reported that the combined probiotic and protein group had a reduced decline in peak power and increased recovery at 24 h and 72 h post-exercise. Such results indicate that probiotic supplementation may enhance the well-known beneficial effects of protein supplementation on muscle recovery and power production, leading to improved performance in sporting domains where power is a primary contributor to performance.

In summary, limited studies have investigated the impact of probiotic intervention on power-specific sporting performance in athletes. Most of these trials have assessed the impact of probiotics in combination with protein, with only one study evaluating the effects of probiotic administration in isolation. Although the preliminary data are promising, future research is needed to determine the isolated impact of probiotics on power performance.

## 4. Probiotic Mechanisms of Actions in Sporting Performance

Whilst the specific mechanisms underpinning the ergogenic benefits of gut microbiome modulation on sporting performance are still to be fully elucidated, several proposed mechanisms have been investigated. Below, we provide an overview of the mechanisms currently being investigated in the field.

### 4.1. Exercise Recovery

The physical demands of training and competition in athletes are high, particularly during intense conditioning and competition preparation. There is a need for rapid recovery from these intense activities to allow for physiological adaptations and enhanced sporting performance. Many interventions have been developed to support recovery from such training regimes, including, but not limited to, cold and heat exposure, mobility, sleep, and medicating with non-steroid anti-inflammatory drugs. However, emerging RCT evidence indicates that gut microbiome modulation may support recovery from intense exercise training.

Lee et al. reported that 6 weeks of supplementation with the probiotic Lactobacillus paracasei significantly reduced blood biomarkers of muscle injury (creatine kinase and myoglobin) and inflammation (C-reactive protein, CRP) following an exercise-induced muscle damage protocol in healthy males and females (aged 20–40 yrs) [10]. The authors also reported a significantly accelerated recovery, subsequently leading to reduced muscle force loss when compared to placebo. In agreement with this trial, another RCT investigated the influence of probiotic supplementation for 17 weeks on muscle soreness and sleep quality in 38 elite rugby union players whilst competing in domestic and international competitions [22]. There was a significant improvement in self-reported muscle soreness and leg heaviness scores in those receiving the probiotic compared to the placebo. Interestingly, the improvements in exercise recovery were associated with improved sleep quality and quantity and decreased CRP biomarker concentrations. Such findings suggest that the well-documented role of gut microbiome modulation in promoting sleep in the general population [23] may also extend to elite athletes during competitions and partly underpin their improved exercise recovery. Furthermore, 21 days of probiotic supplementation in healthy, resistance-trained males led to significantly reduced performance decrements following a muscle-damaging exercise when compared to a placebo [12]. In agreement with the previous trials, significant reductions in inflammatory biomarkers were observed and sustained for up to 48 h post-exercise. In addition, one RCT in healthy trained males reported reductions in inflammatory biomarker response to intense exercise compared to placebo after 14 weeks of probiotic supplementation, although exercise recovery was not explicitly measured [24].

Current evidence indicates that probiotic supplementation may help to support recovery from intense exercise training. This is likely through reductions in the inflammatory response to intense training, whereas some evidence suggests that improved sleep may also underpin the improved recovery rates following probiotic supplementation. However, to date, the evidence is limited and there is a need for future research to investigate optimal dosages, timings, and strains of probiotics to further utilise probiotic supplementation as a tool to improve exercise recovery and subsequently enhance training adaptations and performance.

### 4.2. Nutrient Absorption

The composition of the gut microbiome plays an essential role in the metabolism, absorption, and utilisation of vital nutrients, including vitamins and minerals. One postulated mechanism by which gut microbiome modulation may enhance sporting performance is by increasing the bioavailability of fundamental nutrients essential for sustaining muscle contractions and recovery. For example, branched-chain amino acids (BCAAs: valine, leucine, and isoleucine) play crucial roles in muscular fatigue, and supplementation has been reported to reduce biochemical markers of muscle injury [11]. One trial in trained triathletes found a significant (24–60%) increase in circulating plasma BCAAs following 8-week probiotic supplementation [9]. This increase in BCAAs coincided with significant improvements in exercise performance. In agreement with this finding, the authors of reference [8] conducted an RCT in healthy resistance-trained males to assess the impact of whey protein (20 g) versus whey protein (20 g) combined with probiotics on protein absorption and circulating BCAAs and power performance (one-rep max for leg press and vertical jump). Following the 60-day supplementation period, those assigned to the whey protein and probiotic combined intervention had significantly higher plasma-free amino acids, indicative of greater protein absorption, compared to those assigned to the whey protein alone. In addition, there were significant 34%, 43%, and 32% greater levels of circulating isoleucine, leucine, and valine, respectively, in those consuming the combined whey protein and probiotic compared to the whey protein in isolation. Finally, these improvements in protein absorption were associated with significant increases in power performance, with a 17% and 8% improvement in the one-rep max leg press and vertical jump performance, respectively, in those assigned to the whey protein combined with probiotics.

In addition to improving protein absorption, one RCT investigated the impact of a probiotic intervention on carbohydrate metabolism in trained male cyclists [25]. The authors reported that following the 4-week probiotic intervention, there was a small but significant improvement in the glucose oxidation rate with 60 to 120 min of moderate cycling exercise. Such findings are corroborated by research outside the sports science discipline. Specifically, a recent systematic review and meta-analysis suggested that probiotic intervention in pregnancy may alter carbohydrate metabolism [26]. Finally, there is some preliminary evidence that combined probiotic and iron supplementation in anaemic athletes may enhance the iron status to a greater extent than iron supplementation alone [27].

Probiotic supplementation, particularly in combination with other macronutrients, may improve the absorption, metabolism, and bioavailability of fundamental substrates required to maintain exercise performance, prevent fatigue, and enhance physiological adaptations to training. Specifically, preliminary RCT evidence suggests that probiotics may enhance exercise performance by improving the bioavailability of carbohydrates, protein, and BCAAs, including isoleucine, leucine, and valine.

### 4.3. Gastrointestinal Symptoms

Gastrointestinal symptoms can be defined as upper gastrointestinal symptoms, including nausea, vomiting, and chest pain, and lower gastrointestinal symptoms, such as diarrhoea and flatulence. Gastrointestinal symptoms are widely reported in athletes, particularly in endurance athletes. For example, 27% of recreational runners report moderate or severe gastrointestinal symptoms during a race. The mechanisms underpinning exercise-induced gastrointestinal symptoms are still to be completely elucidated. However, a reduction in the splanchnic blood flow results in the dysregulation of the intestinal barrier, likely leading to an immunological response associated with gastrointestinal symptoms [28]. Additionally, carbohydrate ingestion has also been proposed as another causal factor due to malabsorption when intake is in excess [29].

Probiotics have been shown to alleviate gastrointestinal symptoms in patients with irritable bowel syndrome and support normal bowel movements in the healthy population [2]. More recently, evidence suggests probiotics may be a therapeutic strategy to prevent gastrointestinal symptoms in athletes during intense training and competitions, which may partly underpin the ergogenic benefits of probiotics. Specifically, results from a 90-day intervention with a multistrain probiotic in elite cyclists found significant reductions in gastrointestinal symptoms at rest and during exercise compared to placebo [30]. In addition, those assigned to the probiotic intervention had significantly lower perceived exertion rating post-intervention when compared to placebo. Meanwhile, marathon runners supplemented with a probiotic for 4 weeks reported significantly fewer and less severe gastrointestinal symptoms during training and in a marathon race than those receiving a placebo [12]. Moreover, the authors also highlighted that the participants who were supplemented with probiotics demonstrated reduced performance decrements (as evidenced by maintaining running speed) towards the end of the race compared to those assigned to the placebo.

Emerging evidence from athletes and clinical populations indicates that probiotics may be an effective therapeutic intervention for managing gastrointestinal symptoms. This is an important avenue for future research, considering that athletes are at greater risk of gastrointestinal symptoms than the general population. Future research may investigate the impact of different probiotic strains and the duration of intervention on gastrointestinal symptoms in athletes, as well as the potential ergogenic benefits in real-world sporting scenarios.

### 4.4. Immune Function

Elite athletes undertaking prolonged intense exercise in high-intensity sports such as cycling, swimming, and rowing are at an increased risk of developing upper respiratory tract illness [31]. Thus, the prevention of illness during heavy training regimes and competition is a high priority for coaches and elite athletes. For example, a three-year observational study of over 300 Olympic athletes reported that 70% of the recorded illnesses resulted in time loss (defined as complete absence) from training and competition [32], which can be detrimental to elite performance. In healthy adults, probiotic supplementation has been reported to prevent upper respiratory tract illnesses and episodes of diarrhoea. For example, a study of 721 healthy adults over three winter periods reported a 30–45% reduction in illness episode frequency and duration following a symbiotic intervention [33]. In contrast, a systematic review and meta-analysis of twelve human intervention trials concluded that probiotic supplementation can significantly reduce the frequency of traveller’s diarrhoea [34], a common compliant amongst elite athletes travelling for training and competitions. Although the evidence from healthy adults indicates that probiotics may play a role in preventing illness, which could significantly benefit athletic performance, only a small number of studies have investigated these benefits in athletes specifically, albeit with promising results. For example, one RCT in 20 elite distance runners reported significant reductions in upper respiratory tract illness frequency and severity after 28 days of probiotic supplementation [35]. Another RCT found significant reductions in upper respiratory tract illness following a four-month probiotic intervention during a winter training period in university athletes [36].

The preliminary evidence to date suggests that probiotic supplementation in athletes may help to prevent illness owing to upper respiratory tract infections, particularly during prolonged exercise regimes or during competitions. However, further RCTs are required to confirm these benefits.

### 4.5. Performance Anxiety

High-intensity exercise over prolonged periods, such as preseason and during competitions, has been associated with increased anxiety and stress. Such mood states can be detrimental to sporting performance and further the frequency/intensity of training preparations. It is well documented that probiotic intervention can induce anxiolytic effects in the general population [2]. For example, in an 8-week probiotic RCT conducted at our centre, we reported significant reductions in stress and improved mood states in healthy adults (Jarrett et al., in preparation). Therefore, probiotic supplementation may represent a cost-effective and easily accessible intervention to support mental health and well-being in elite and recreational athletes. Indeed, the anxiolytic effects of probiotic supplementation have been investigated in competitive badminton players. Specifically, a 6-week RCT with the probiotic strain Lactobacillus Casei Shirota for 6 weeks resulted in a significant 16% and 20% reduction in anxiety and stress levels, respectively, whilst no change was reported in the placebo group [4]. In addition, a study in competitive male football players reported a significant increase in both delta and theta brain waves following 4-week supplementation with Lactobacillus Casei Shirota compared to that with placebo, indicative of greater attention and relaxation [37]. Indeed, the authors also reported significant improvements in sustained attention in the active probiotic group compared to placebo.

To date, preliminary evidence has suggested that probiotic supplementation may have critical anxiolytic benefits in athletes who are predisposed to increased stress and anxiety. Such an effect could have essential benefits for the mental well-being and sporting performance of athletes. However, only a small number of trials have been conducted that are explicitly focused on team sports (football and badminton). Therefore, the preliminary results should be taken with caution, and future research is needed to confirm these findings. Future research should also consider the effect of probiotics on mental well-being and attention in solo sports, such as archery or table tennis, which require sustained attention.

## 5. Conclusions

It is important to acknowledge the strengths and limitations of this review. The main strength of this review is the assessment of the effects of gut microbiome modulation through probiotics on power-specific sporting domains, an area which has, to date, been widely overlooked. Moreover, this review provides a detailed breakdown of the potential underpinning mechanisms by which gut microbiome modulation may impact athletic performance, providing avenues for future research. There are a number of limitations of this review. The nature of this narrative review means no systematic search of the literature was conducted and it is possible that some human trials were potentially not included. Moreover, the limited number of trials investigating the impact of gut microbiome modulation on athletic performance limited the ability to address important factors that may influence supplement efficacy, including the probiotic strain and formulation and individual factors such as the baseline dietary intake and gut microbiome composition.

Evidence indicates that gut microbiome modulation may significantly impact sporting performance, particularly in endurance domains. Most studies to date have investigated the efficacy of probiotics, with the most substantial evidence supporting the use of a multistrain probiotic intervention to improve endurance performance. Future research may wish to examine the effectiveness of a combined prebiotic and probiotic (synbiotic) on sporting performance. Moreover, preliminary evidence indicates that probiotic interventions may also enhance power-specific sporting domains; however, this requires further research. Whilst other parameters beyond the endurance and power domains impact sporting performance, to date, there has been insufficient evidence to review these additional parameters. The mechanisms are still to be fully elucidated; however, preliminary research suggests that increased aerobic capacity may partly underpin improvements in endurance performance. Additional evidence indicates that improved exercise recovery and reduced GI symptoms and illness during strenuous training and competition may further enhance training and performance capacity. Finally, limited evidence also suggests that gut microbiome modulation may enhance sporting performance through improving nutrient absorption and utilisation, particularly concerning carbohydrates, protein, and BCAAs.

In conclusion, the current evidence indicates that probiotic supplementation may benefit athletes’ sporting performance, particularly in respect to the endurance domains.

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
