# Peer review of "The Role of the Gut Microbiome and Probiotics in Sports Performance: A Narrative Review Update"

_nutrients, 2025, doi:10.3390/nu17040690_

Round 1

Reviewer 1 Report

Comments and Suggestions for Authors

The type of review has to be mentioned in the title, in the abstract, and in the whole manuscript.

The abstract is not proper. It should be structured in Background/Objectives; Methods (the applied methodologies such as searched databases, inclusion/exclusion criteria, and keywords used in the search need to be provided); Results; Conclusions, and future perspectives.

The keywords presented by the authors are scarce. More keywords related to the carried-out study should be provided.

Lines 29-30 and 31-34: References are missing.

The section 1 (Introduction) is very poor. Much more is expected from an introductory section of a review manuscript submitted to an international Q1 journal like Nutrients. More citations and more detailed data about the topics to be addressed must be given; The rationale and justification to carry out the study have to be clarified, as well as its novelty.

A Methods section is missing. See the example of section 2 of this published paper: https://www.mdpi.com/2304-8158/10/6/1175

The section called “Discussion” is inappropriate. This title should be replaced by another one more suitable. “Discussion” is what the authors should make in the whole manuscript and the title of this section should be direct to the addressed topics.

Tables are missing in the whole manuscript with the summary of the reviewed studies and their highlights.

All the subsections in this “Discussion” section are poorly analyzed and much more is expected from a review manuscript. For example, how is possible to have only about 10 lines to discuss the interaction between the gut microbiome and exercise? There are a lot of investigations you should include here. The same for the other parts of this “Discussion”.

Based on this, it is obvious that Conclusions also need to be improved and aligned with the revised/new manuscript.

Author Response

We would like to thank reviewer 1 for their insightful comments and feedback. However, we believe this review provides a timely and important update on the role of probiotic supplementation on athletic performance, as indicated by reviewer 3.

We have reviewed the comments and have made revisions to the original manuscript based upon the reviewers comments. Please see below a detailed breakdown of the amendments:

Comment 1:

  • "The type of review has to be mentioned in the title, in the abstract, and in the whole manuscript."

 Response:

  • The title has been amended to "The role of the gut microbiome and probiotics in sports performance: A Narrative Review Update"- line 2-3
  • Reference to the narrative review has been made in the abstract- line 12

Comment 2:

  • The abstract is not proper. It should be structured in Background/Objectives; Methods (the applied methodologies such as searched databases, inclusion/exclusion criteria, and keywords used in the search need to be provided); Results; Conclusions, and future perspectives.

 Response:

  • The abstract has been amended to include these headings, with the exception of explicit "methods", as per the review design, the nature of the review is "narrative" and does not incorporate a systematic search of the literature, the manuscript was an invited review from the journal editors. Line 10-25

Comment 3:

  • "The keywords presented by the authors are scarce. More keywords related to the carried-out study should be provided."

 Response:

  • The keywords provided encompass the scope of the review. We have however, added 2 additional keywords- "Nutrient Absorption" and Immune System", which also encompass the mechanisms discussed throughout the review Line 27.

Comment 4:

  • Lines 29-30 and 31-34: References are missing.

Response:

  • Thank you for highlighting this, the references have now been added- Line 36

Comment 5:

  • Introduction- "More citations and more detailed data about the topics to be addressed must be given; The rationale and justification to carry out the study have to be clarified, as well as its novelty"

Response:

  • Additional citations, references to literature in specific sporting domain responses to probiotic interventions and potential mechanisms provided. Rationale and justification for the study clarified and novelty of the review explicitly stated. 

Comment 6:

  • "A Methods section is missing."

Response

  • Thank you as this is an invited narrative review, a methods section, we do not deem necessary. Since a systematic review was not conducted a methods section would not be applicable here. As per previous invited narrative reviews published in Nutrients, which do not include a specific methods section. See narrative review published by Gao et al (2013), titled: The role of Probiotics in Skin Health and Related Gut-Skin Axis: A review. https://www.mdpi.com/2072-6643/15/14/3123

Comment 7:

  • "The section called “Discussion” is inappropriate"

Response

  • We have removed the title heading !Discussion" and renamed the title headings throughout

Comment 8:

  • Discussion- "how is possible to have only about 10 lines to discuss the interaction between the gut microbiome and exercise? There are a lot of investigations you should include here"

Response

  • We agree there is more literature on this topic, however, a brief overview pertaining to the influence of exercise on gut microbiome composition was provided, since this is not within the specific focus of the invited narrative review. The focus is upon the impact of probiotics on gut microbiome diversity and how this subsequently impacts upon sporting performance and thus we did not want to detract form the primary focus and message of this invited article. 

Comment 9:

  • Conclusions also need to be improved and aligned with the revised/new manuscript

Response

  • Conclusion has been updated in line with revised manuscript- Lines 320-338.

Reviewer 2 Report

Comments and Suggestions for Authors

My recommendations are the following:
Abstract recommend mentioning the specific aspects regarding the Methods section.
Lines 31-34 recommend mentioning the bibliographic sources.
Lines 43-46 recommend mentioning the bibliographic sources.
Lines 55-56 mention - Several observational studies- but there is only one study mentioned in the bibliography, I recommend clarification.
Introductions- I recommend expanding by making concrete arguments regarding the topic of this study. Also for a review-type article, mentioning only two bibliographic indexes is too little and does not support the ideas. Also, you present an article in preparation, I recommend deleting it, even if the name of the first author is mentioned.
The Interaction Between the Gut Microbiome and Exercise- I recommend expanding it, taking into consideration other physical activities on the topic of the subsection.
Lines 84-87 recommend deleting it, an article that has not been published cannot be mentioned.
Index 6 is not mentioned correctly in the bibliography, part of the title is missing, I recommend correction.
Lines 107-111 recommend mentioning the bibliographical sources.
Lines 148-152 recommend mentioning the bibliographical sources.
Exercise Recovery: - mention two bibliographical sources after which in lines 172-175 you present a supporting conclusion, this aspect is too poorly outlined, I recommend revision based on scientific evidence.
Lines 228-230 the statement is not supported, I recommend clarifications.
In conclusion, the bibliographical indices mentioned are insufficient. The subsections presented also do not support the variety of sports and physical activities clearly targeting microbial influences as supplements to support efforts.
This article is not specific to a review, the scientific bases for selecting the articles were not mentioned; the keywords are too few; a PRISMA diagram was not created; the conclusions mentioned in each subsection are not sufficiently supported. The presentation of aspects regarding endurance and physical strength are poorly outlined. Sport does not consist only of these two qualities. I recommend a major revision to the authors before resubmitting.

Author Response

We would like to thank reviewer 2 for their insightful comments and feedback. We have addressed their suggestions below:

Comment 1:

  • Abstract recommend mentioning the specific aspects regarding the Methods section

 Response:

  • Thank you, as this is an invited narrative review, a methods section, we do not deem necessary. Since a systematic review was not conducted a methods section would not be applicable here. As per previous invited narrative reviews published in Nutrients, which do not include a specific methods section. See narrative review published by Gao et al (2013), titled: The role of Probiotics in Skin Health and Related Gut-Skin Axis: A review. https://www.mdpi.com/2072-6643/15/14/3123

Comments 2, 3 & 4:

  • "Lines 31-34 recommend mentioning the bibliographic sources.
    Lines 43-46 recommend mentioning the bibliographic sources.
    Lines 55-56 mention - Several observational studies- but there is only one study mentioned in the bibliography, I recommend clarification."

 Response:

  • References added to line 36, lines 50-58 and additional observational study added to line 76

Comment 5:

  • Introduction- "expanding by making concrete arguments regarding the topic of this study" and "Also, you present an article in preparation, I recommend deleting it,"

 Response:

  • The introduction has been expanded with additional references and details provided- Lines 47-70
  • Reference to the article in press has been removed, since the preceding sentences discusses results from a systematic review of 10 studies, no additional reference was deemed necessary

Comment 6:

  • "Lines 84-87 recommend deleting it, an article that has not been published cannot be mentioned."

Response:

  • Thank you, we have removed this reference- Line 105

Comment 7:

  • "Index 6 is not mentioned correctly in the bibliography, part of the title is missing, I recommend correction."

Response:

  • Thank you or highlighting, this has now been amended- Line 369 

Comment 8 and 9:

  • "Lines 107-111 recommend mentioning the bibliographical sources.
    Lines 148-152 recommend mentioning the bibliographical sources."

Response:

  • References added to line 186 and 198

Comment 10:

  • "Exercise Recovery: - mention two bibliographical sources after which in lines 172-175 you present a supporting conclusion"

Response:

  • Thank you for this comment. The lack of  literature discussed reflects the limited available evidence to date. We have included an additional RCT which did not look at exercise recovery specifically but does report inflammatory biomarker response to probiotic intervention- Line 191. The conclusions for this section have been amended to reflect the paucity of the currently available data. Line 198-199

Comment 11:

  • Lines 228-230 the statement is not supported, I recommend clarifications.

Response:

  • We have provided clarifications on this statement. Line 256

Comment 12:

  • "This article is not specific to a review, the scientific bases for selecting the articles were not mentioned"

Responses:

  • Thank you as this is an invited narrative review, a methods section, we do not deem necessary. Since a systematic review was not conducted a methods section would not be applicable here. As per previous invited narrative reviews published in Nutrients, which do not include a specific methods section. See narrative review published by Gao et al (2013), titled: The role of Probiotics in Skin Health and Related Gut-Skin Axis: A review. https://www.mdpi.com/2072-6643/15/14/3123
  • The title has been amended to reflect the type of review "The role of the gut microbiome and probiotics in sports performance: A Narrative Review Update"- line 2-3
  • Reference to the narrative review has been made in the abstract- line 12

Comment 13:

  • "The keywords are too few"

Responses:

  • The keywords provided encompass the scope of the review. We have however, added 2 additional keywords- "Nutrient Absorption" and Immune System", which also encompass the mechanisms discussed throughout the review Line 27.

Comment 14:

  • "The presentation of aspects regarding endurance and physical strength are poorly outlined. Sport does not consist only of these two qualities"

Response:

  • We agree that these are not the only 2 domains that impact upon sporting performance. As mentioned in line- "Research to date has primarily focused on two specific aspects of sporting performance concerning the ergogenic effects of probiotic supplementation: endurance and power performance"- There is limited evidence for other aspects and thus we could not review other domains. We have provided clarity in the conclusion - Lines 334-336

Reviewer 3 Report

Comments and Suggestions for Authors

This work reviews gut microbiome modulation through probiotics and its effects on exercise performance.

The exploration of probiotics about exercise performance is timely and relevant, given the increasing interest in gut health and its systemic implications. By acknowledging the need for further research, particularly in specific performance domains like power performance and the mechanisms of action, the review sets a foundation for future studies.

The assertion that probiotic supplementation improves endurance performance is not strongly supported by a large body of trials. The limited number of studies, especially for power performance, raises questions about the generalizability of the findings.

While the review mentions potential mechanisms (e.g., recovery, nutrient absorption), it fails to delve into the biological processes by which probiotics exert their effects. Without this, the claims remain somewhat speculative.

The mention of probiotics potentially alleviating performance anxiety is intriguing but lacks robust evidence. This area appears under-researched and should be approached with caution until more substantial data are available.

The manuscript does not account for the variability in probiotic strains and formulations, which may lead to differing effects on performance. A more nuanced discussion of this could enhance the understanding of probiotic efficacy.

The review does not adequately address how individual differences, such as diet, genetics, and baseline gut microbiome composition, may influence the effectiveness of probiotic supplementation.

Author Response

We would like to thank reviewer 1 for their insightful comments and feedback and agree that the article provides a timely exploration of probiotics impact upon exercise performance.

Comment 1:

  • "The assertion that probiotic supplementation improves endurance performance is not strongly supported by a large body of trials. The limited number of studies, especially for power performance, raises questions about the generalizability of the findings."

 Response:

We agree with the reviewers comments, we have included the literature that is available to date but we acknowledge the need for further research. With this review article we aim to highlight areas for future research including the need for further investigations into the role of probiotics on power performance and an elucidation of the mechanisms underpinning any observed benefits to both endurance and power domains. We have amended the conclusions based upon the reviewers comments and thank them for their comments.  Lines 326 and 331

Comment 2:

  • While the review mentions potential mechanisms (e.g., recovery, nutrient absorption), it fails to delve into the biological processes by which probiotics exert their effects

 Response:

Thank you for this insightful comment. We agree that an exploration of the biological and molecular processes that underpin the potential mechanisms would be of interest and provide further information to the reader. However, the focus of this invited narrative review is to provide an overview of the current state of the evidence in relation to human trials of gut microbiome modulation and sporting performance with a secondary focus on potential mechanisms explored. Whilst the inclusion of biological and molecular mechanistic underpinnings would be of interest, we feel it would takeaway from the key primary outcome and falls out of scope for this narrative review.  

Comment 3:

  • The mention of probiotics potentially alleviating performance anxiety is intriguing but lacks robust evidence. This area appears under-researched and should be approached with caution until more substantial data are available.

 Response:

We agree, there is some evidence for the effects of probiotics on anxiety in the general population and specific patient groups. However, the research in the sporting domain is indeed extremely limited, to the two RCTs discussed in the article. To reflect this, we have added additional information and caution in the interpretation of the anxiolytic effects of probiotics, specifically applied to the sporting domain- Lines 322-324.

Comment 4:

  • The manuscript does not account for the variability in probiotic strains and formulations, which may lead to differing effects on performance. A more nuanced discussion of this could enhance the understanding of probiotic efficacy.

 Response:

We acknowledge that different probiotic strains and formulations may impact supplement efficacy. However, the limited available evidence meant it was not possible to assess specific strain efficacy, due to the various different strains, dosages, intervention periods and populations used across the different interventions. It is difficult to disentangle the impact of probiotic strains Vs different study characteristics, such as study duration, the intervention population, the specific athletic assessment used etc. We have acknowledged this limitation in the review conclusion. Line 328-339.

Comment 5:

  • The review does not adequately address how individual differences, such as diet, genetics, and baseline gut microbiome composition, may influence the effectiveness of probiotic supplementation.

 Response:

We agree, due to the limited available evidence and the lack of reporting of such parameters in the published articles reviewed, we were unable to account for such individual differences. We have now acknowledged this limitation within the conclusion. Lines 328-339

Round 2

Reviewer 1 Report

Comments and Suggestions for Authors

A Methods section is missing. See the example of section 2 of this published paper: https://www.mdpi.com/2304-8158/10/6/1175. The fact that this is a narrative review does not mean that the methodologies applied do not have to be described, as demonstrated by the example of the indicated paper. This should be addressed in the abstract and the manuscript with the inclusion of a new section.

The abstract is not well structured. Background/Objectives should be the same subsection, followed by Methods, Results, and Conclusions/Future Perspectives.

I still believe the authors should better discuss the topic related to the interaction between the gut microbiome and exercise.

Author Response

Comment 1:

  • "A Methods section is missing. See the example of section 2 of this published paper: https://www.mdpi.com/2304-8158/10/6/1175. The fact that this is a narrative review does not mean that the methodologies applied do not have to be described, as demonstrated by the example of the indicated paper. This should be addressed in the abstract and the manuscript with the inclusion of a new section."

Response:

We thank the reviewer for providing this comment. However, we believe a "methods" section for this narrative review is unwarranted. Nutrients journal does not provide guidance for the structure of a narrative review and is at the discretion of the invited author. We thank the reviewer for providing the reference, however, this example, although published in an MDPI journal, this is the journal "Foods" not Nutrients. We would like to draw the reviewers attention to a number of narrative reviews recently published in this journal Nutrients, whereby no Methods section was included in the abstracts nor manuscripts:

Role of Dietary Nutrients in the Modulation of Gut Microbiota: A Narrative Review- https://www.mdpi.com/2072-6643/12/2/381

Nutrients and Dietary Approaches in Patients with Type 2 Diabetes Mellitus and Cardiovascular Disease: A Narrative Review- https://www.mdpi.com/2072-6643/13/11/4150

The Influence of Nutrition on Adiponectin—A Narrative Review- https://www.mdpi.com/2072-6643/13/5/1394

Nutrition in Menopausal Women: A Narrative Review- https://www.mdpi.com/2072-6643/13/7/2149

Nutritional Considerations for Peripheral Arterial Disease: A Narrative Review- https://www.mdpi.com/2072-6643/11/6/1219

Nutrition in the Intensive Care Unit—A Narrative Review- https://www.mdpi.com/2072-6643/13/8/2851

We hope this provides clarity on this issue but if further information is needed, the Nutrients Journal Editor will be able to provide further information. 

Comment 2:

  • "The abstract is not well structured. Background/Objectives should be the same subsection, followed by Methods, Results, and Conclusions/Future Perspectives."

Response:

Thank you for your input. The abstract structure has now been amended, with the exception of an inclusion of a methods section (see above response 1). Lines- 11, 15, 17-18 and 24.

Comment 3:

  • "I still believe the authors should better discuss the topic related to the interaction between the gut microbiome and exercise."

Response:

We thank the reviewer for their comments. We have added additional background and references to this section. Lines- 69-73. 79-80 and 81.

Reviewer 2 Report

Comments and Suggestions for Authors

no comments

Author Response

We would like to take the opportunity to thank the reviewer for their time in providing insightful comments which have helped to improve the quality of this manuscript. 

Reviewer 3 Report

Comments and Suggestions for Authors

Considering the major changes made by the authors and all the answers given, I believe it can be published.

Author Response

(The authors gave the same response as above.)
